# Care for post-COVID-19 condition in Germany from the perspectives of patients, informal caregivers and general practitioners: Study protocol for a mixed methods study

Melanie Brinkmann[1‡]*, Maike Stolz[2,3‡]*, Annika Herr[3,4], Christoph Herrmann-Lingen[5], Imke Koch[1], Christiane Müller[6], Frank Müller[6,7], Uta Sekanina[6], Jona Theodor Stahmeyer[8], Martina de Zwaan[9‡], Christian Krauth[2,3‡], Nils Schneider[1‡]

1 Institute for General Practice and Palliative Care, Hannover Medical School, Hannover, Germany, 2 Institute for Epidemiology, Social Medicine and Health Systems Research, Hannover Medical School, Hannover, Germany, 3 Center for Health Economics Research (CHERH), Hannover, Germany, 4 Institute of Health Economics, Leibniz University Hannover, Hannover, Germany, 5 Department of Psychosomatic Medicine and Psychotherapy, University Medical Center Göttingen, Göttingen, Germany, 6 Department of General Practice, University Medical Center Göttingen, Göttingen, Germany, 7 Department of Family Medicine, Michigan State University, College of Human Medicine, Michigan, East Lansing, United States of America, 8 Health Services Research Unit, AOK Niedersachsen, Hannover, Germany, 9 Department of Psychosomatic Medicine and Psychotherapy, Hannover Medical School, Hannover, Germany

‡ MB and MS are contributed equally to this work and share first authorship. MZ, CK and NS are authors contributed equally to this work and share last authorship.
* stolz.maike@mh-hannover.de (MS); brinkmann.melanie2@mh-hannover.de (MB)

**Data Availability Statement:** No datasets were generated or analysed during the current study. All

## Abstract

### Background

A large number of individuals suffer from post-COVID-19 condition (PCC), characterised by persistent symptoms following a SARS-CoV-2 infection with an impact on daily personal and professional activities. This study aims at examining which (health) care services are used by PCC patients in the German federal state of Lower Saxony, and how these patients manage their condition. The perspectives of patients, informal caregivers and general practitioners (GPs) will be considered.

### Methods

The study will employ a mixed methods design. Patients' perspective will be evaluated through an online survey of: (1) 21,000 adult individuals with a PCC diagnosis (ICD10 U09.9!) in their statutory health insurance claims data in 2022 ("AOK survey") and (2) a self-selected sample of adult individuals with a proven SARS-CoV-2 infection in 2023 and persistent symptoms ("public survey"). Additional data sources will be claims data (n = 27,275) and 25–30 semi-structured interviews. Informal caregivers, perspective will be collected through an online survey and semi-structured interviews. GPs' perspective will be evaluated through four focus groups involving six to eight participants each and an online survey of all registered and practicing GPs in Lower Saxony (approximately 5,000). All survey data will be descriptively analysed. In addition, correlation analyses and multivariable regression

relevant data from this study will be made available upon study completion.

**Funding:** The present study is supported by the COVID-19-Research Network Lower Saxony (COFONI), through funding from the Ministry of Science and Culture of Lower Saxony in Germany (14-76403-184). The funder played no role in the conceptualisation, design, data collection, analysis, decision to publish or preparation of the manuscript.

**Competing interests:** CH-L has received a lecture honorarium from Novartis and royalties from Hogrefe Publish-ers for the German version of the Hospital Anxiety and Depression Scale. His research is supported by grants from the German Ministry of Education and Research, the German Re-search Foundation and the European Commission. This does not alter our adherence to PLOS ONE policies on sharing data and materials. The other authors declare that they have no competing interests.

analyses will be conducted, for example on factors influencing affected individuals' use of medical services. Interview and focus group data will be subjected to qualitative content analysis. A health economic analysis will be used to determine the costs of PCC to health care payers, patients and society. The project will conclude with an expert workshop to dis-cuss the results and derive recommendations.

## Discussion

The results of the study will provide a multidimensional description of the (health) care situa-tion and needs of patients with PCC, and derive recommendations for improving health care.

## Trial registration

The VePoKaP study is registered at the German Clinical Trials Register (DRKS00032846).

## Introduction

In recent years, the SARS-CoV-2 pandemic has placed global health care systems and socie-ties under tremendous pressure. A COVID-19 disease can have far-reaching consequences, including symptoms that persist well beyond the acute infection and affect not only persons who were severely ill, but also those who had only mild symptoms initially [1]. The WHO defines these persistent complaints as a post-COVID-19 condition (PCC), which "occurs in individuals with a history of probable or confirmed SARS-CoV-2 infection, usually 3 months from the onset of COVID-19 with symptoms that last for at least 2 months and cannot be explained by an alternative diagnosis. Common symptoms include fatigue, shortness of breath, and cognitive dysfunction, and generally have an impact on everyday functioning" [2]. PCC, and especially fatigue as a frequent occurring symptom of PCC, has a significant impact on individuals' physical and mental wellbeing, thereby affecting their daily personal and professional activities [3].

Estimates of PCC prevalence vary widely, ranging from 2.3% to 16% for non-hospitalised patients [4]. However, it is currently being discussed that methodological flaws have led to a significant overestimation of PCC prevalence in many scientific publications [5, 6]. Notwith-standing this potential overestimation, the number of individuals affected by PCC is likely high, given the exceptionally high number of people who have been infected with SARS-CoV-2. Thus, even considering a low prevalence rate, PCC represents a significant burden on health care systems and societies, worldwide [7].

A study in the USA found that the direct medical costs associated with PCC in the first 6 months of persistent symptoms (i.e., starting 31 days after SARS-CoV-2 infection) were 1.46 times higher than those for controls with no equivalent SARS-CoV-2 infection [8]. Addition-ally, PCC is associated with indirect costs due to productivity losses—either through time off work, presentism or disability. In Germany alone, a study projected the overall economic, health care and pension costs associated with PCC with an onset in 2021 at approximately 7.4 billion euros [9]. Furthermore, PCC gives rise to social and societal costs due to the burden that is placed on relatives and friends. The functional limitations associated with PCC impact patients' social lives and fulfilment of everyday tasks [3, 4], thereby necessitating the support services of informal caregivers (non-professional persons (both kin and non-kin caregivers

like neighbours and friends) who provide care in a home setting). This aspect has not yet been sufficiently addressed in the literature.

General practitioners (GPs) are typically the first point of contact for patients with persistent symptoms following a SARS-CoV-2 infection. Thus, they play a key role in the management of PCC patients [10–12]. Although PCC is characterised by certain common symptoms, its clinical presentation remains extremely heterogeneous [13]. Accordingly, GPs report substantial uncertainty in their diagnosis and treatment of the condition [11, 12, 14]. As other possible causes for the symptoms must be ruled out, it is often the case that patients are assessed by different medical specialists, making the diagnosis and treatment process very time-consuming (for both the affected individuals and GPs) [12, 14]. The high level of suffering experienced by patients, combined with a lack of well-established and effective therapies, can also lead patients to visit many different medical specialists to try new therapeutic approaches [15]. Such approaches may include conventional medical treatments as well as complementary and alternative medical procedures. Little is known about the health care services utilised by PCC patients and their pathways through the health care system. As a proportion of affected individuals do not appear to fully recover over the long term [16, 17], it is important to evaluate how they manage their symptoms and complaints.

The present protocol describes the study "Care for patients with post-COVID: Analysis of claims data and the perspectives of patients, informal caregivers and general practitioners (VePoKaP)" [German: Versorgung von Patient*innen mit Post-COVID: Analysen von Kassendaten und von Perspektiven der Patient*innen, Angehörigen und Hausärzt*innen (VePoKaP)] registered at the German Clinical Trials Register (DRKS00032846). The study findings will be used to derive recommendations for improving the care of individuals affected by PCC in Lower Saxony, a German federal state comprised of rural, urban and metropolitan regions. Additionally, as both hospital planning and outpatient care management are conducted at the federal state level, the federal state approach of the present study may best support a successful science-practice-policy transfer.

## Objectives

The study aims at evaluating individuals affected or formerly affected by PCC in Lower Saxony, with respect to their use of health care services and management of their complaints. In doing so, the research will consider the perspectives of individuals at different stages of PCC and with different levels of PCC severity, as well as those of informal caregivers and GPs.

The following research questions will be answered:

- Perspective of affected and formerly affected individuals

1) Which health care services are used by PCC patients? To what extent are complementary and alternative services used? Are there regional differences in Lower Saxony (comparison of rural, urban and metropolitan regions)? How satisfied are PCC patients with the health care services available to them? Which health care services are lacking?

2) What strategies do affected individuals use to cope with PCC at work and in everyday life? How important is support from informal caregivers?

3) Is there an association between the severity of PCC and the ways in which affected individuals cope with their symptoms, the ways in which affected individuals use health care services, or the needs of affected individuals?

- Perspective of informal caregivers

4) How do informal caregivers experience and cope with the disease? What is their caregiver burden and what support do they need?

- Perspective of GPs

5) How do GPs rate the health care of individuals with PCC and their use of health care ser-vices in terms of the diagnostic process, availability and quality of treatment options and improvement in health status? How do GPs perceive individuals with PCC to be coping with the disease? What do GPs identify as the health care needs of individuals with PCC?

- Health economic perspective

6) What are the costs of PCC from the perspective of health care payers, patients and the soci-ety? What is the economic impact of PCC on the affected persons and their informal caregivers?

## Materials and methods

The study employs a mixed methods design, integrating quantitative and qualitative methods of health services research, secondary data analysis and health economic analysis (Fig 1).

### Data collection

**Perspective of affected and formerly affected individuals.** The perspectives of individu-als affected or formerly affected by PCC will be assessed through two separate cross-sectional online surveys and an evaluation of claims data from the largest statutory health insurance com-pany in Lower Saxony (AOK Lower Saxony), which covers one-third of the adult population (2,370,541/6,676,314) [18]. In addition, semi-structured interviews will be conducted (Fig 1).

For the first online survey (AOK survey), adult individuals meeting the following inclusion criteria will be invited by post to participate in the research: (a) insured with AOK Lower

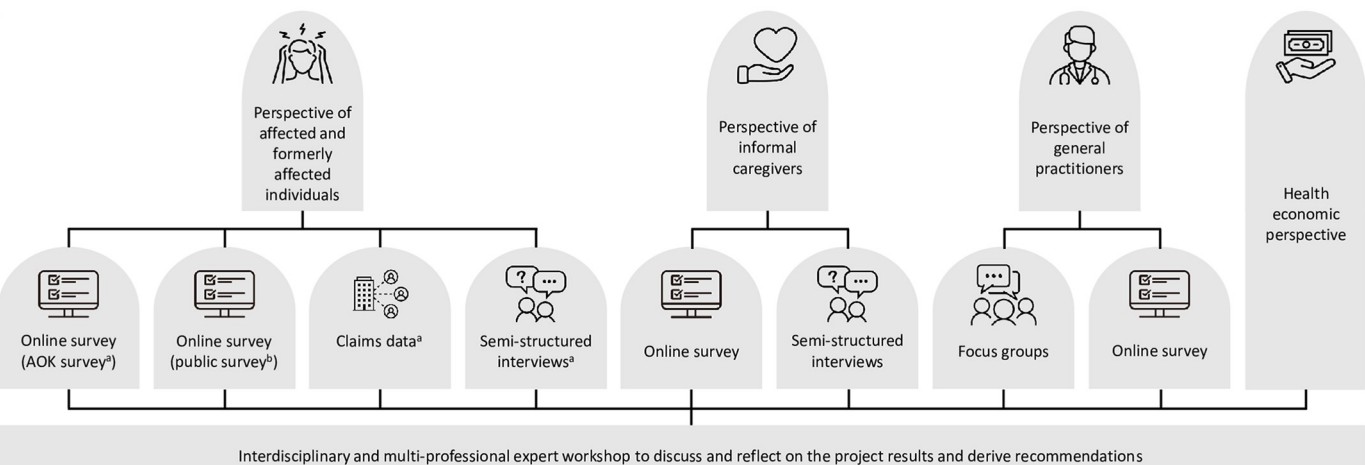

**Fig 1. Study design: Perspectives and data collection methods.** © Hanna A. A. Röwer via Canva.com. [a] individuals who have statutory health insurance with the AOK Lower Saxony; ≥ 18 years old; post-COVID-19 condition (ICD10 U09.9) in their outpatient billing data or certificates of incapacity for work (claims data) in 2022; place of residence: Lower Saxony; continuously insured with the AOK Lower Saxony since 2019; no individuals whose affairs are managed by a guardian; no employees of the AOK Lower Saxony. [b] Persons with SARS-CoV-2 infection (self-test sufficient) in 2023 and persistent symptoms in self-perception; ≥ 18 years old; place of residence: Lower Saxony.

Saxony, (b) diagnosed with PCC (ICD10 U09.9) in their claims data (outpatient billing data or a certificate of incapacity for work in 2022), (c) resident of Lower Saxony and (d) continuously insured with AOK Lower Saxony since 2019. Individuals whose affairs are managed by a guardian or who are employees of the health insurance company will be excluded, resulting in a final sample of 27,275 (AOK claims data group). For budgetary reasons, a random sample of 21,000 insured persons will be selected and included (AOK survey group). The invitation will contain a link to the online questionnaire and a randomly assigned individual identification number (ID), serving as an access code. The ID will also be included in the AOK Lower Saxony claims data, to create a personal link between the primary online survey data and the second-ary claims data for the analyses. To maximise the response rate, a reminder letter will be sent to individuals who have not completed the survey after 2 weeks. In addition, telephone support will be offered to clarify open questions and solve technical problems. Assuming a response rate of 25%, $n$ = 5,000 participants are expected.

Given the criterion of a PCC diagnosis in 2022, the AOK survey sample is likely to include both individuals who remain affected by PCC at the time of the survey completion (affected individuals) and individuals who have already recovered (formerly affected individuals). In order to be able to address the participants appropriately, the survey will begin with the ques-tion of whether the person still has symptoms or whether they have already recovered. Depending on the answer, the following questions are then asked in a customised form with the help of a filter guide. For example: "How often are you currently restricted in carrying out your normal everyday tasks (e.g. household, grocery shopping, childcare,. . .) due to your post-COVID symptoms?" vs. "How often were you restricted in carrying out your normal everyday tasks (e.g. household, grocery shopping, childcare,. . .) due to your post-COVID symptoms?" The reference year of 2022 was chosen because outpatient billing data in Germany are only available from statutory health insurance funds with a time lag of about 9 months. However, this reference year will also ensure that the findings of the AOK survey on the development of symptoms, the need for treatment and the utilisation of medical services refer to a significant period of time.

The selection of patients for the AOK survey is based on a medical PCC diagnosis, which is generally considered trustworthy. However, it is also possible to validate this secondary data against the primary data from the AOK survey to ensure that only people who have had a pre-vious SARS-CoV-2 infection and whose symptoms have persisted for three months according to the WHO definition are actually included in the analyses.

To recruit participants for the second online survey (public survey), AOK Lower Saxony will launch a public call via its social media channels and member magazine. Eligible partici-pants meet the following criteria: (a) tested positive for SARS-CoV-2 in 2023 (self-test suffi-cient), (b) report persistent symptoms and (c) reside in Lower Saxony (group 2). Interested individuals will be able to access the online survey via a link or a QR code. This public survey group recruitment via self-selection will aim at reaching currently affected individuals who have not necessarily presented to outpatient or inpatient care, or have not been diagnosed with PCC by a GP or medical specialist. Despite the bias associated with self-reporting the presence of PCC symptoms, this potentially vulnerable group is considered separately. Individual ques-tions are added to validate the self-report.

Both online surveys will include: (a) self-developed items referring to sociodemographic and socioeconomic factors, symptoms and their progression, utilisation of health care services (including complementary and alternative medical services), satisfaction with care and impair-ments in everyday and working life (S1 Appendix); and (b) standardised instruments (Table 1). Answering the survey is estimated to take about 30 minutes.

Table 1. Standardised instruments included in both online surveys of affected individuals.

| Patient-related constructs and outcomes | Instruments |
|---|---|
| Fatigue | Chalder Fatigue Scale [19, 20] |
| Post-exertional malaise | DePaul Symptom Questionnaire—Post-Exertional-Malaise (DSQ-PEM) [21, 22] |
| Anxiety and depression | Patient Health Questionnaire (PHQ-4) [23] |
| Health-related quality of life | EQ-5D-5L [24, 25] |
| Social support | Brief Social Support Scale (BS-6) [26] |
| Perception of social participation | Short Scale Measuring Perceived Social Participation (KsT-5) [27] |
| Self-efficacy | Short form of the General Self-Efficacy Scale (GSE-6) [28] |
| Coping with illness | Short form of the Freiburger Fragebogen zur Krankheitsverarbeitung (FKV-15) [29] |

The comprehensibility of the questionnaire will be informally verified through pretesting with colleagues and affected individuals, who complete the online survey and give feedback on each question [30].

Currently affected respondents who indicate a recovery time of 14 hours or more in the DSQ-PEM will be offered the option of psychosomatic counselling at Hannover Medical School (MHH). The goal is to make sure that these severely affected patients receive the necessary help and to support them in finding adequate treatment if needed. The number of respondents who make use of this counselling service, and the specific needs they report, will be documented.

The claims data of the 27,275 adult individuals with a PCC diagnosis in 2022 who are insured by AOK Lower Saxony (AOK claims data group) will serve as a secondary data source. The claims data of the statutory health insurers contain, in addition to the insured persons' sociodemographic data, billing data from reimbursed healthcare services providing an overview of the services utilised by patients from the time of diagnosis, with respect to outpatient care, medication, therapeutics, inpatient care and rehabilitation, and periods of sick leave incurred. In order to chart at least 6 months of data follow-up after diagnosis, the first three quarters of 2023 will be taken into account, in addition to the year 2022. Pre-existing conditions or comorbidities will be identified in the preceding years of 2019–2021. Information on the settlement structure of the place of residence (e.g., urban/rural area) will be obtained on the basis of the classification of the Federal Institute for Research on Building, Urban Affairs and Spatial Development [31]. Where relevant, claims data will be linked to the primary survey data for any insured individual who participate in the AOK online survey.

In addition to the two online surveys and claims data, semi-structured interviews will be conducted to collect further perspectives from affected and formerly affected individuals. Eligible participants for these interviews include adult individuals with statutory health insurance who are able to participate in an interview regarding their health status in the German language. Participants will be purposefully sampled based on characteristics such as health status (i.e., affected/formerly affected), sex, age, place of residence (i.e., urban/rural) and presence/absence of informal support. This criterion sampling will allow to consider different perspectives [32–34]. An interview invitation leaflet will be sent to all statutory insured individuals of AOK Lower Saxony, alongside the invitation to the AOK online survey. Anticipating high willingness to participate, only a random sample of 25% of the total number of insured individuals contacted for the AOK online survey will be invited. Individuals who are interested in participating in the interviews will be invited to contact the research team by telephone or

email. A total number of 25–30 interviews will be conducted [35, 36]. In preparation for the interviews, an interview guide on topics such as individual strategies for coping with fatigue related to PCC, experiences with primary care, met and unmet needs, and wishes for health care will be developed on the basis of the literature and discussions within the interdisciplinary research team (S3 Appendix). Prior to data collection, the interview guide will be piloted and refined, where necessary. To enable a description of the sample, interview participants will also complete a short questionnaire focused on sociodemographic factors. Half of the interviews will be conducted by a researcher from MHH and the other half by a researcher from University Medical Center Göttingen (UMG). The 30- to 60-minute interviews will be audio-recorded and conducted at MHH, UMG or participants' homes, or via video conference or telephone.

**Perspective of informal caregivers.** The perspective of informal caregivers will be determined via a cross-sectional online survey and semi-structured interviews (Fig 1). Informal caregivers will be recruited via the affected or formerly affected individuals in group 1. An additional invitation letter, which can be passed on to relevant caregivers, accompanies the invitation letter from AOK Lower Saxony. The online survey data of the affected or formerly affected individuals (group 1) and informal caregivers may be linked via an ID (subject to the patient's consent).

The survey will comprise: (a) self-developed items on participants' sociodemographic and socioeconomic factors, as well as the type of support offered and time required for these tasks (S2 Appendix); and (b) standardised instruments (Table 2). Answering the survey is estimated to take about 15 minutes.

The comprehensibility of the questionnaire will be informally pretested with colleagues, who complete the online survey and give feedback on each item [30].

The perspective of informal caregivers will also be captured through the semi-structured interviews with affected or formerly affected individuals, who will be invited to determine whether they would like their caregiver to take part in the interview with them. This offer is made on the assumption that individuals who would otherwise decline participation in an interview may be more willing to participate in the presence of a familiar person. Eligible informal caregivers must be ≥ 18 years old and the primary confidant of the affected or formerly affected individual, and they must provide care or emotional support to them. Questions addressing informal caregivers will be included in the interview guide for the interviews with affected or formerly affected individuals.

**Perspective of GPs.** GPs' perspective will be evaluated sequentially, through focus groups and an online survey (Fig 1). For the focus groups, potential participants (all registered GPs) will be recruited by invitation letter and by telephone, via the teaching and research networks of the Institute for General Practice and Palliative Care of MHH and the Department of General Practice at UMG. GPs who treat or have treated people with PCC and are interested in taking part will be invited to contact the research team by telephone or email. To enable diverse

**Table 2. Standardised instruments included in the online survey of informal caregivers.**

| Informal caregiver-related constructs and outcomes | Instruments |
|---|---|
| Impact of informal care | Care-Related Quality of Life of Caregivers (Carer-QoL-7D) [37, 38] |
| Anxiety and depression | Patient Health Questionnaire (PHQ-4) [23] |
| Health-related quality of life | EQ-5D-5L [24, 25] |
| Satisfaction with life | Satisfaction with Life Scale (SWLS) [39, 40] |

perspectives within and across the focus groups, participants will be purposefully selected using criterion sampling based on characteristics such as residence (i.e., urban/rural) and type of practice (i.e., working alone/in a partnership) [32–34]. In total, four focus groups, two in each centre, will be conducted to gain an insight into urban (MHH) and more rural (UMG) areas, each involving six to eight participants [41] to promote a good discussion atmosphere in which every participant can express themselves. The interview guide will include questions about how affected individuals cope with the disease; the role of GPs in caring for affected individuals; and GPs' wishes, expectations and needs. In addition, each participant will be asked to complete a short survey comprised of self-developed items addressing sociodemographic factors. Two focus groups will be conducted by researchers from MHH and the remaining focus groups will be conducted by researchers from UMG. Each focus group will be moderated by two researchers and will last up to 90 minutes. All focus groups will be audio-recorded, and they will be conducted at MHH, UMG or, if necessary, via video conference. Participants will be compensated 100€.

For the online survey, all registered and practicing GPs in Lower Saxony, except those treating only privately insured patients, will be eligible (approximately $n = 5,000$). GPs will be invited by fax or by email to participate. The invitation letter will include a QR code and link to the online survey, which will be developed on the basis of the results of the previous focus groups. It is expected that the survey will address topics such as the health care experiences, needs and expectations of affected individuals and how affected individuals cope with the disease (from GPs' own perspectives). Assuming a response rate of 15–20%, $n = 1,000$ participants are expected. To maximise the response rate, a reminder will be sent to GPs who have not completed the online survey 2 weeks after the first invitation. In addition, the survey will be kept as short as possible [42].

## Data analysis

Data from the online surveys with affected and formerly affected individuals, informal caregivers and GPs will be analysed descriptively. First, bivariate statistics will be used to analyse the patient data, with the aim of determining associations between patient-related constructs and outcomes and utilisation of medical services. In addition, the association between PCC symptom severity and informal caregiver burden will be determined. Second, multivariable regression models will be applied to determine relevant influencing factors for the utilisation of medical services.

The audio-recorded interviews with affected and formerly affected individuals and informal caregivers, and the focus groups with GPs, will be transcribed verbatim and analysed using qualitative content analysis [43–45]. Data analysis will be conducted using MAXQDA (VERBI Software, Berlin, Germany).

As part of the health economic analysis (Fig 1), PCC costs will be determined from the perspectives of health care payers, patients and society. The monetary valuation of services and expenses will follow standard valuation procedures [46, 47], if not already shown in the secondary claims data. Costs will be shown both in total and by area (i.e., outpatient care, medication, therapeutics, inpatient care, rehabilitation, alternative/complementary medical care).

## Interdisciplinary and multiprofessional expert workshop

The study will conclude with an interdisciplinary and multiprofessional expert workshop, aimed at discussing the quantitative and qualitative results and deriving recommendations, thereby enabling a successful science-practice-policy transfer. Experts in PCC from different perspectives and with different responsibilities (e.g., the Long COVID Expert Council of the

Lower Saxony Ministry of Science and Culture; experts from medicine, rehabilitation, public health, epidemiology, self-government, politics and self-help) will be invited to participate. The half-day workshop will include both small group work and plenaries, moderated by members of the research team. The results of the workshop will be documented and synthesised.

### Ethical considerations

Ethics approval was obtained from the Ethics Committees of Hannover Medical School (reference number 11077_BO_K_2023) and University Medical Center Göttingen (reference number 15/10/23 Ü). The study will be conducted in accordance with the principles of the Declaration of Helsinki.

All potential participants are informed that their participation in the study is voluntary and they have the right to refuse or withdraw at any time without any disadvantage. Participation in the study is based on written (i.e., interviews with affected and formerly affected individuals and informal caregivers, and focus groups with GPs) or electronic (i.e., online surveys of affected and formerly affected individuals, informal caregivers and GPs) informed consent. The data will be deleted ten years after the end of the project.

## Discussion

The results of the study will provide a multidimensional perspective on the health care situation and needs of individuals with PCC, thereby providing a basis for recommendations to improve health services. The focus on the federal state of Lower Saxony will allow for multidimensional analyses and the refinement of regional health policy, for the first time (as far as we are aware) in Germany.

The study comes with some limitations. First, a response bias determined by a difference between responders and non-responders is likely. The results of other surveys show, for example, that women tend to be overrepresented in the responder group [48, 49], while data for age are inconsistent [48]. It can also be postulated that individuals with severe symptoms and a high disease burden will be more likely to participate in the study than individuals who are no longer or only slightly affected by PCC. Differences between responders and non-responders may also manifest within the informal caregiver and GP groups, with those with greater burden more likely to participate. To evaluate this risk of bias, a non-responder-analysis will be performed for the AOK survey and the GP online survey data. For the latter, however, a comparison will only be possible with regard to sex and practice location. To increase the response rates for the online surveys (i.e., of affected and formerly affected individuals in the AOK survey, informal caregivers and GPs) and thus reduce the risk of response bias, all invitation letters are written in simple language and include contact information for the research team. In addition, reminders are sent to those who have not yet completed the surveys 2 weeks after the initial invitation [42].

Second, the retrospective assessment of, for example, the time of the SARS-CoV-2 infection (which is presumed to be the cause of PCC) and the severity of PCC symptoms or symptom management may be limited by recall bias. It is also possible that some participants may be unable to reliably assess whether the symptoms they associate with PCC occurred for the first time after their SARS-CoV-2 infection or if they had already been present before. While the risk of bias can be assumed to be consistent across all survey samples, it may differ across affected and formerly affected individuals.

Lastly, individuals who are insured through AOK Lower Saxony differ from the general population in Lower Saxony (and Germany, more broadly) in terms of age distribution, vocational education qualifications, complexity of occupation and occupational field [50, 51],

which may limit the generalisability of the results. Lower Saxony in general comprises nearly 10% of the German population and is made up of rural, urban and metropolitan regions. In terms of age structure and unemployment rate, Lower Saxony is very close to the national average. The proportion of foreigners, the proportion of the employed persons with academic qualifications and the net income per capita are slightly lower than the German average [52]. However, due to the rather small differences, the results of this study are expected to be transferable to the whole of Germany. Strengths of the study include its mixed methods design, enabling a deeper understanding of individual perception. Together with the analysis of statutory health insurance claims data, the approach has the potential to compensate for the methodological limitations of individual data collection and analysis. Another strength is the consideration of different perspectives, allowing the health care situation and needs (for change) to be described in a multidimensional way. Finally, the multidisciplinary expert workshop, where the project results will be discussed by relevant stakeholders and recommendations will be derived, will support the translation of the research findings into practice.

## Supporting information

**S1 Appendix. Self-developed items included in both online surveys of affected individuals.**
(PDF)

**S2 Appendix. Self-developed items included in the online survey of informal caregivers.**
(PDF)

**S3 Appendix. Guidelines for semi-structured interviews with PCC patients and their informal caregivers.**
(PDF)

**S1 Checklist. STROBE Statement—Checklist of items that should be included in reports of observational studies.**
(PDF)

## Acknowledgments

We thank Hanna A. A. Röwer from the Institute for General Practice and Palliative Care at Hannover Medical School for her support with the graphical visualisation of the study design.

## Author Contributions

**Conceptualization:** Melanie Brinkmann, Maike Stolz, Christoph Herrmann-Lingen, Imke Koch, Christiane Müller, Frank Müller, Uta Sekanina, Jona Theodor Stahmeyer, Martina de Zwaan, Christian Krauth, Nils Schneider.

**Funding acquisition:** Melanie Brinkmann, Maike Stolz, Christoph Herrmann-Lingen, Christiane Müller, Frank Müller, Jona Theodor Stahmeyer, Martina de Zwaan, Christian Krauth, Nils Schneider.

**Methodology:** Melanie Brinkmann, Maike Stolz, Annika Herr, Christoph Herrmann-Lingen, Christiane Müller, Frank Müller, Jona Theodor Stahmeyer, Martina de Zwaan, Christian Krauth, Nils Schneider.

**Project administration:** Christoph Herrmann-Lingen, Frank Müller, Jona Theodor Stahmeyer, Martina de Zwaan, Christian Krauth, Nils Schneider.

**Supervision:** Christian Krauth, Nils Schneider.

**Writing – original draft:** Melanie Brinkmann, Maike Stolz.

**Writing – review & editing:** Melanie Brinkmann, Maike Stolz, Annika Herr, Christoph Herr-mann-Lingen, Christiane Müller, Frank Müller, Jona Theodor Stahmeyer, Martina de Zwaan, Christian Krauth, Nils Schneider.

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
