## [Decision Letter · Decision Letter 0]

30 Jul 2024

PONE-D-24-12578Care for post-COVID-19 condition in Germany from the perspectives of patients, informal caregivers and general practitioners: Study protocol for a mixed methods studyPLOS ONE

Dear Dr. Stolz,

Thank you for submitting your manuscript to PLOS ONE. After careful consideration, we feel that it has merit but does not fully meet PLOS ONE’s publication criteria as it currently stands. Therefore, we invite you to submit a revised version of the manuscript that addresses the points raised during the review process.

Please note that we have only been able to secure a single reviewer to assess your manuscript. We are issuing a decision on your manuscript at this point to prevent further delays in the evaluation of your manuscript. Please be aware that the editor who handles your revised manuscript might find it necessary to invite additional reviewers to assess this work once the revised manuscript is submitted. However, we will aim to proceed on the basis of this single review if possible. 

We look forward to receiving your revised manuscript.

Kind regards,

Vanessa Carels

Staff Editor

PLOS ONE

“CH-L has received a lecture honorarium from Novartis and royalties from Hogrefe Publishers for the German version of the Hospital Anxiety and Depression Scale. His research is support-ed by grants from the German Ministry of Education and Research, the German Research Foundation and the European Commission. The other authors declare that they have no com-peting interests.”

Please include your updated Competing Interests statement in your cover letter; we will change the online submission form on your behalf."

3. In the online submission form, you indicated that [The survey data used and analysed in this study can be obtained in anonymised form from the study team, upon reasonable request.].

Reviewers' comments:

Reviewer's Responses to Questions

**Comments to the Author**

1. Does the manuscript provide a valid rationale for the proposed study, with clearly identified and justified research questions?

Reviewer #1: Partly

2. Is the protocol technically sound and planned in a manner that will lead to a meaningful outcome and allow testing the stated hypotheses?

Reviewer #1: Partly

3. Is the methodology feasible and described in sufficient detail to allow the work to be replicable?

Reviewer #1: No

4. Have the authors described where all data underlying the findings will be made available when the study is complete?

Reviewer #1: Yes

5. Is the manuscript presented in an intelligible fashion and written in standard English?

Reviewer #1: No

6. Review Comments to the Author

You may also provide optional suggestions and comments to authors that they might find helpful in planning their study.

Reviewer #1: Two general comments; How FS fit in with the PCS needs to be better described.

All surveys and interview schedules should be provided as appendices.

Abstract:

The comment about “fatigue syndrome is one of the most common symptoms of PCC” appears a bit ‘random’ in the abstract since the following sentence is about PCC more broadly. I would recommended either adding more context – e.g., tell the reader you’re paying particular attention to fatigue syndrome. Or remove it from the abstract.

Methods:

the tense of this section needs to be changed to future since it’s a protocol. E..g, Informal caregivers perspective will be collected rather than informal caregivers perspective is collected”

What is AOK?

It is unlcear what these two surveys are. It appears as though you are describing two cohorts of participants rather than actual surveys. This should be clarified.

What does the GP online survey evaluate/collect?

There is no information on what is collected in the survey data therefore it’s unclear if descriptive analysis is appropriate.

Who are the “experts” involved in the workshops.

Introduction:

Line 73: It is very important that “fatigue syndrome (FS)” be carefully defined. The authors should include what syndrome they are describing here. For review see: Sandler CX, Wyller VBB, Moss-Morris R, Buchwald D, Crawley E, Hautvast J, Katz BZ, Knoop H, Little P, Taylor R, Wensaas KA, Lloyd AR. Long COVID and Post-infective Fatigue Syndrome: A Review. Open Forum Infect Dis. 2021 Sep 9;8(10):ofab440. doi: 10.1093/ofid/ofab440. PMID: 34631916; PMCID: PMC8496765.

What is meant by “claims data” (line 108). Is this private health insurance claim. Please note the reader may not be familiar with the German health care system and may require some background information to understand the context.

It may be helpful to provide some context for Lower Saxony and how this area of Germany is different/similar to the rest of the country. Highlighting this may provide a stronger rationale for the benefits of this study. (note, I have made this comment in the discussion section too).

Objectives:

This is the first time the reader has understood PCC to be different to FS. The authors should clarify this. Wouldn’t there be people who have “both”

Why is only the perspective of GPs included. What about other specialist e.g., ID physicians, respiratory physicians etc, nurses

What is mean by complementary and alternative services? Do you mean complimentary services?

How will you assess if there are regional differences if you don’t assess any other region?

Informal caregivers should be defined. What’s the difference between formal and informal. And why wouldn’t you include formal?

Line 138 – how can GPs “rate the health care of individuals? Are you referring to the quality of health care AVAILABLE to individuals?

Methods:

Line 150: Figure 1 is very difficult to read – this needs to be improved. Claims data needs to be defined. AOK needs to be defined

Line 164 – change to “will be “ conducted

Line 167 – how will the reliability of the PCC diagnosis be assessed. As the authors stated in the introduction – the PCC diagnosis requires careful medical and psychiatric investigation. Is there data available to demonstrate this has happened. Will all patient have confirmed COVID-19? What definition of PCC will be used? It would be helpful to reader to have this included in the text

Regarding the second online survey – if they have not been diagnosed with PCC or there is only self-selection of COVID-19 – what is the benefit of including them? The biases related to this cohort should be described.

One suggestion would be to give each of these survey’s a name (e.g., AOK survey for survey 1 and public survey for survey 2) to help the reader identify the difference in cohorts.

Line 193: could the authors include the details of the in-house demographic questions collected. E.g. similar to table 1 so the reader has a full access to the survey. Alternatively the survey could be provided as an appendix.

I want to congratulate the authors on selecting well-validated and relevant questionnaires.

What are the potential bias by selecting those insured by AOK only?

Line 199 – how long is it anticipated the survey will take. This information should be included in the methods.

Line 203: what does psychosomatic counselling at Hannover Medical School (MHH) involve and why is it important for participants to be offered this?

Line 206 – how is group 3 different to group 1?

The authors should consider adding subheadings to the methods to help the reader understand the groups and methods used.

The semi-structured interview scheduled should be made available as an appendix. Same for the care givers.

What are the criteria for the GPs? Do they need to have an interest/speciality in PCS?

How was the sample size of the GP focus groups determined?

Data analysis:

How will “formally affected” individuals be defined? Is this self-nominated? Is it based on level of function or work status? This detail should also be added to the methods.

What is the sample size of the workshop? What will be the outcome?

Discussion:

The authors should include some detail on although this study is conducted in a specific area of Germany – how can it inform the wider literature. It may also be helpful to provide some context for Lower Saxony and how this area of Germany is different/similar to the rest of the country. Highlighting this may provide a stronger rationale for the benefits of this study.

7. PLOS authors have the option to publish the peer review history of their article (what does this mean?). If published, this will include your full peer review and any attached files.

Reviewer #1: No

---

## [Author Response · Author response to Decision Letter 0]

10 Sep 2024

Dear editor, We would like to thank the reviewer for the useful comments. Please find below our answers to the comments of the reviewer and journal requirements. We have revised our manuscript accordingly. Kind regards, Maike Stolz, on behalf of all authors.

We have reviewed the requirements and made appropriate adjustments.

“CH-L has received a lecture honorarium from Novartis and royalties from Hogrefe Publishers for the German version of the Hospital Anxiety and Depression Scale. His research is supported by grants from the German Ministry of Education and Research, the German Research Foundation and the European Commission. The other authors declare that they have no competing interests.”

Please include your updated Competing Interests statement in your cover letter; we will change the online submission form on your behalf."

We have added the corresponding sentence.

3. In the online submission form, you indicated that [The survey data used and analysed in this study can be obtained in anonymised form from the study team, upon reasonable request.].

We have adapted the statement accordingly.

“No data is generated or analysed for the current publication. Relevant data will be made available via an online repository when the study is completed and published in peer reviewed journals. To maintain patient privacy, the transcripts from the qualitative data as well as claims data sets will be only made available upon reasonable request after approval from research ethics board.”

We have checked the reference list but have not found any necessary adjustments except the newly added source based on the reviewer's comments.

Reviewer comments

Reviewer #1: Two general comments; How FS fit in with the PCS needs to be better described. All surveys and interview schedules should be provided as appendices.

Thank you very much for your comprehensive review and helpful suggestions.

Unfortunately, at this stage of the project, we cannot provide all interview guidelines and surveys, as some of them are still being developed in the course of the project (this mainly concerns the surveys of GPs). Due to the online format of the surveys with detailed filtering, it is also not possible to make the entire surveys available as a document. However, as suggested as an alternative, we have summarised all the contents of the questionnaires in detail in a table and made them available as S1-S3 appendices.

Please find our further adaptations and responses below.

Abstract: The comment about “fatigue syndrome is one of the most common symptoms of PCC” appears a bit ‘random’ in the abstract since the following sentence is about PCC more broadly. I would recommended either adding more context – e.g., tell the reader you’re paying particular attention to fatigue syndrome. Or remove it from the abstract.

Thank you for your suggestion. We removed this part from the manuscript.

Methods: the tense of this section needs to be changed to future since it’s a protocol. E..g, Informal caregivers perspective will be collected rather than informal caregivers perspective is collected”

Thank you for your remark. We have changed it accordingly in the whole manuscript.

What is AOK?

The AOK Lower Saxony is a statutory health insurance fund in Lower Saxony. We removed this abbreviation from the abstract because it is not relevant here.

It is unclear what these two surveys are. It appears as though you are describing two cohorts of participants rather than actual surveys. This should be clarified.

Thank you for this recommendation. We have reworded the part of the summary to make it clearer.

Line 43-48: “Patients’ perspective will be evaluated through an online survey of: (1) 21,000 adult individuals with a PCC diagnosis (ICD10 U09.9!) in their statutory health insurance

claims data in 2022 (“AOK survey”) and (2) a self-selected sample of adult individuals with a proven SARS-CoV-2 infection in 2023 and persistent symptoms (“public survey”).”

What does the GP online survey evaluate/collect? There is no information on what is collected in the survey data therefore it’s unclear if descriptive analysis is appropriate. Who are the “experts” involved in the workshops.

We understand that it would be helpful at this point to have more information about the content of the surveys and the methods used. However, due to the limited number of characters available for the abstract, it is not possible to include all of this additional information.

Introduction: Line 73: It is very important that “fatigue syndrome (FS)” be carefully defined. The authors should include what syndrome they are describing here. For review see: Sandler CX, Wyller VBB, Moss-Morris R, Buchwald D, Crawley E, Hautvast J, Katz BZ, Knoop H, Little P, Taylor R, Wensaas KA, Lloyd AR. Long COVID and Post-infective Fatigue Syndrome: A Review. Open Forum Infect Dis. 2021 Sep 9;8(10):ofab440. doi: 10.1093/ofid/ofab440. PMID: 34631916; PMCID: PMC8496765.

Thank you for pointing this out. After reviewing the recommended source, we decided not to put a special emphasis on "fatigue syndrome", but to mention fatigue (measured on the Chalder Fatigue Scale) as one of the most common symptoms of PCC, as we have no way to specify and validate "post-COVID fatigue" with the collected data in the recommended way. We have changed the entire manuscript accordingly, especially in the "Objectives" section (line 134-152).

What is meant by “claims data” (line 108). Is this private health insurance claim. Please note the reader may not be familiar with the German health care system and may require some background information to understand the context.

We understand the concerns. However, this is the title under which the study was registered in the German Clinical Trials Registry, so we have not changed it in the manuscript. We have added the registration information for clarification of the context.

Line 112-116: “The present protocol describes the study “Care for patients with post-COVID: Analysis of claims data and the perspectives of patients, informal caregivers and general practitioners (VePoKaP)” [German: Versorgung von Patient*innen mit Post-COVID: Analysen von Kassendaten und von Perspektiven der Patient*innen, Angehörigen und Hausärzt*innen (VePoKaP)] registered at the German Clinical Trials Register (DRKS00032846).”

We have also reworded the part on claims data in the abstract to make it clearer.

Line 44-46: “(1) 21,000 adult individuals with a PCC diagnosis (ICD10 U09.9!) in their statutory health insurance claims data in 2022 (“AOK survey”)”

It may be helpful to provide some context for Lower Saxony and how this area of Germany is different/similar to the rest of the country. Highlighting this may provide a stronger rationale for the benefits of this study. (note, I have made this comment in the discussion section too). We have added additional information mainly in the discussion. Line 406-412: “Lower Saxony in general comprises nearly 10 % of the German population and is made up of rural, urban and metropolitan regions. In terms of age structure and unemployment rate, Lower Saxony is very close to the national average. The proportion of foreigners, the proportion of the employed persons with academic qualifications and the net income per capita are slightly lower than the German average [52]. However, due to the rather small differences, the results of this study are expected to be transferable to the whole of Germany.”

Objectives: This is the first time the reader has understood PCC to be different to FS. The authors should clarify this. Wouldn’t there be people who have “both”

Thank you for your advice. We have changed the manuscript as described above.

Why is only the perspective of GPs included. What about other specialist e.g., ID physicians, respiratory physicians etc, nurses

You are right that the perspectives of other professionals would also be interesting. Due to limited resources, we had to decide which professional perspective we wanted to include. We opted for the perspective of GPs, as they are usually the first point of contact in the healthcare system, the first to diagnose PCC and play a key role in the management of PCC patients and their multiplicity of symptoms.

What is mean by complementary and alternative services? Do you mean complimentary services?

“Complementary and alternative medical procedures” is a common term for non-conventional treatment methods (for review see e.g.: 10.3389/fnut.2021.791899 ) like natural medicine or traditional Chinese medicine.

How will you assess if there are regional differences if you don’t assess any other region?

Lower Saxony comprises rural, urban and metropolitan regions which are to be compared with each other. We have added it accordingly.

Line 130-132: 1) Which health care services are used by PCC patients? To what extent are complementary and alternative services used? Are there regional differences in Lower Saxony (comparison rural, urban and metropolitan regions)?

Informal caregivers should be defined. What’s the difference between formal and informal. And why wouldn’t you include formal?

Informal caregivers are non-professional persons (such as a family member, neighbour or friend) who provide care in a home setting for another person (for review see e.g.: https://doi.org/10.1111/inr.12194). We included this information in the introduction. Line 92-95: “The functional limitations associated with PCC impact patients’ social lives and fulfilment of everyday tasks [3, 4], thereby necessitating the support services of informal caregivers (non-professional persons (both kin and non-kin caregivers like neighbours and friends) who provide care in a home setting).” Formal carers, on the other hand, are healthcare professionals and are included here in the form of general practitioners. Line 138 – how can GPs “rate the health care of individuals? Are you referring to the quality of health care AVAILABLE to individuals?

Thank you for this question. We have reworded the question to make our intention clearer. Line 144-146: “How do GPs rate the health care of individuals with PCC in terms of the diagnostic process, availability and quality of treatment options and improvement in health status?” Methods: Line 150: Figure 1 is very difficult to read – this needs to be improved. Claims data needs to be defined. AOK needs to be defined

Thank you for your advice. We have uploaded our figure file to the PACE diagnostic tool to make sure that it meets PLOS ONE requirements and adapted the explanation to make it clearer.

Line 158-165:” Fig 1. Study design: Perspectives and data collection methods. © Hanna A. A. Röwer via Canva.com. a individuals who have statutory health insurance with the AOK Lower Saxony; ≥ 18 years old; post-COVID-19 condition (ICD10 U09.9) in their outpatient billing data or certificates of incapacity for work (claims data) in 2022; place of residence: Lower Saxony; continuously insured with the AOK Lower Saxony since 2019; no individuals whose affairs are managed by a guardian; no employees of the AOK Lower Saxony. b Persons with SARS-CoV-2 infection (self-test sufficient) in 2023 and persistent symptoms in self-perception; ≥ 18 years old; place of residence: Lower Saxony.”

Line 164 – change to “will be “ conducted

Thank you for your remark. We have adjusted the tense throughout the methods section.

Line 167 – how will the reliability of the PCC diagnosis be assessed. As the authors stated in the introduction – the PCC diagnosis requires careful medical and psychiatric investigation. Is there data available to demonstrate this has happened. Will all patient have confirmed COVID-19? What definition of PCC will be used? It would be helpful to reader to have this included in the text

The validity of the PCC diagnosis is indeed a point that needs to be carefully discussed especially in subsequent publications of the results. In principle, we trust the medical diagnosis.

The explanation of the diagnosis code ICD U09.9! states: "This code number is to be used when an otherwise classified disease related to a previous coronavirus disease-2019 (COVID-19) is to be reported. This code number is not to be used if COVID-19 is still present.", so we assume that all those affected had a SARS-CoV-2 infection. However, the online survey also asks again how long the symptoms have been present and whether a SARS-CoV-2 infection was present, so that we can validate the information from the claims data and only include patients in the analyses who were still affected 12 or more weeks after an infection according to the WHO definition of PCC used. At the same time, it is possible to see from the claims data which medical specialists were consulted due to the PCC diagnosis. We have included a section on this issue in the methods.

Line 206-210: “The selection of patients for the AOK survey is based on a medical PCC diagnosis, which is generally considered trustworthy. However, it is also possible to validate this secondary data against the primary data from the AOK survey to ensure that only individuals who have had a previous SARS-CoV-2 infection and whose symptoms have persisted for three months according to the WHO definition are actually included in the analyses.“

Regarding the second online survey – if they have not been diagnosed with PCC or there is only self-selection of COVID-19 – what is the benefit of including them? The biases related to this cohort should be described.

The self-selected sample was integrated in order to reach people who had not yet seen a GP or specialist or had not received a PCC diagnosis. This was done on the assumption that this group could differ from the AOK sample, e.g. with regard to the utilisation of alternative medical measures which are not included in the claims data. There are some additional questions asked to validate the self-report. We added this information in the methods section.

Line 219-224: “This public survey group recruitment via self-selection will aim at reaching currently affe

---

## [Decision Letter · Decision Letter 1]

4 Oct 2024

PONE-D-24-12578R1Care for post-COVID-19 condition in Germany from the perspectives of patients, informal caregivers and general practitioners: Study protocol for a mixed methods studyPLOS ONE

Dear Dr. Stolz,

Thank you for submitting your manuscript to PLOS ONE. After careful consideration, we feel that it has merit but does not fully meet PLOS ONE’s publication criteria as it currently stands. Therefore, we invite you to submit a revised version of the manuscript that addresses the points raised during the review process.

We look forward to receiving your revised manuscript.

Kind regards,

Mr. Nuru Abdu

Academic Editor

PLOS ONE

Journal Requirements:

Reviewers' comments:

Reviewer's Responses to Questions

**Comments to the Author**

1. Does the manuscript provide a valid rationale for the proposed study, with clearly identified and justified research questions?

Reviewer #1: Yes

Reviewer #2: Yes

2. Is the protocol technically sound and planned in a manner that will lead to a meaningful outcome and allow testing the stated hypotheses?

Reviewer #1: Yes

Reviewer #2: Yes

3. Is the methodology feasible and described in sufficient detail to allow the work to be replicable?

Reviewer #1: No

Reviewer #2: Yes

4. Have the authors described where all data underlying the findings will be made available when the study is complete?

Reviewer #1: Yes

Reviewer #2: No

5. Is the manuscript presented in an intelligible fashion and written in standard English?

Reviewer #1: Yes

Reviewer #2: Yes

6. Review Comments to the Author

You may also provide optional suggestions and comments to authors that they might find helpful in planning their study.

Reviewer #1: Thank you for considering my remarks. I feel that have been adequately addressed.

i have no further comments.

Reviewer #2: The study is interesting and the use of mixed method to reduce confounders and biases makes the protocol original. However, there are some issues that should be in addressed:

1. Define informal health care givers and the scope of the GPs to be included in the survey.

2. When referring PCC in the objectives section use the formal definition used in the introduction section of the protocol. Why FS, as it has different ICD and will confound the results.

3. Why are the second survey group selected as self-selection, as this will lead to bias. What other strategies did you use to reduce bias other than the DSQ-PEM.

4. The claims data should be clarified, as readers may not be familiar with the system

5. Provide the survey questionnaire contents to be used for GPs.

6. Provide justification how the survey conducted in Lower Saxony can be generalizable to Germany, or provide as limitation in the discussion section.

7. The expected duration of the survey is not included and should be provided.

7. PLOS authors have the option to publish the peer review history of their article (what does this mean?). If published, this will include your full peer review and any attached files.

Reviewer #1: No

Reviewer #2: No

---

## [Author Response · Author response to Decision Letter 1]

21 Oct 2024

Dear editor,

We would like to thank reviewer 1 for the positive review and reviewer 2 for the useful comments. We feel that the comments of reviewer 2 refer to the manuscript originally submitted in March rather than the revised version submitted in September, as almost all of the comments have already been addressed in the revised version on the basis of very similar comments from reviewer 1. Nevertheless, we have taken the helpful suggestions of reviewer 2 into account and made an addition in lines 244-248. 

Please find below our answers to the comments of the reviewer 2.

Kind regards,

Maike Stolz, on behalf of all authors.

Reviewer comments

Reviewer #1: Thank you for considering my remarks. I feel that have been adequately addressed.

i have no further comments.

Thank you for the positive review.

Reviewer #2: The study is interesting and the use of mixed method to reduce confounders and biases makes the protocol original. However, there are some issues that should be in addressed:

Thank you very much for your comprehensive review and helpful suggestions. 

We feel that your comments refer to the manuscript originally submitted in March rather than a revised version submitted in September, as almost all of your comments have already been addressed in this version on the basis of very similar comments from a second reviewer.

Please find the corresponding changes below.

1. Define informal health care givers and the scope of the GPs to be included in the survey.

We included the definition of informal caregivers in the introduction. 

Line 92-95: “The functional limitations associated with PCC impact patients’ social lives and fulfilment of everyday tasks [3, 4], thereby necessitating the support services of informal caregivers (non-professional persons (both kin and non-kin caregivers like neighbours and friends) who provide care in a home setting).”

We are not quite sure what is meant by ‘scope of the GPs included’. To be invited to the survey, the only requirement is that you are registered as a GP with a license from the statutory health insurance provider and are actively practicing. To participate in the preparatory focus groups, the GPs must also have treated patients with PCC. We have added this accordingly.

Line 310-311: GPs who treat or have treated people with PCC and are interested in taking part will be invited to contact the research team by telephone or email.

2. When referring PCC in the objectives section use the formal definition used in the introduction section of the protocol. Why FS, as it has different ICD and will confound the results.

Thank you for pointing this out. We will now not put a special emphasis on "fatigue syndrome", but mention fatigue (measured on the Chalder Fatigue Scale) as one of the most common symptoms of PCC. We have changed the entire manuscript accordingly, especially in the "Objectives" section (line 134-152).

3. Why are the second survey group selected as self-selection, as this will lead to bias. What other strategies did you use to reduce bias other than the DSQ-PEM.

The self-selected sample was integrated in order to reach people who had not yet seen a GP or specialist or had not received a PCC diagnosis. This was done on the assumption that this group could differ from the AOK sample, e.g. with regard to the utilisation of alternative medical measures which are not included in the claims data. There are some additional questions asked to validate the self-report. We added this information in the methods section.

Line 219-224: “This public survey group recruitment via self-selection will aim at reaching currently affected individuals who have not necessarily presented to outpatient or inpatient care, or have not been diagnosed with PCC by a GP or medical specialist. Despite the bias associated with self-reporting the presence of PCC symptoms, this potentially vulnerable group is considered separately. Individual additional questions are asked to validate the self-report.”

4. The claims data should be clarified, as readers may not be familiar with the system

We have reworded the part on claims data in the abstract and added a sentence in the methods section to make it clearer. 

Line 44-46: “(1) 21,000 adult individuals with a PCC diagnosis (ICD10 U09.9!) in their statutory health insurance claims data in 2022 (“AOK survey”)”

Line 242-248: “The claims data of the 27,275 adult individuals with a PCC diagnosis in 2022 who are insured by AOK Lower Saxony (AOK claims data group) will serve as a secondary data source. The routine data of the statutory health insurers contain, in addition to the insured persons' sociodemographic data, billing data from reimbursed healthcare services providing an overview of the services utilised by patients from the time of diagnosis, with respect to outpatient care, medication, therapeutics, inpatient care and rehabilitation, and periods of sick leave incurred.”

5. Provide the survey questionnaire contents to be used for GPs.

Unfortunately, we cannot provide the content of the survey for the general practitioners at this stage of the project, as it is still being identified based on the results of the focus groups.

6. Provide justification how the survey conducted in Lower Saxony can be generalizable to Germany, or provide as limitation in the discussion section.

Thank you for this advice. We have added additional information to the discussion.

Line 408-414: “Lower Saxony in general comprises nearly 10 % of the German population and is made up of rural, urban and metropolitan regions. In terms of age structure and unemployment rate, Lower Saxony is very close to the national average. The proportion of foreigners, the proportion of the employed persons with academic qualifications and the net income per capita are slightly lower than the German average [52]. However, due to the rather small differences, the results of this study are expected to be transferable to the whole of Germany.”

7. The expected duration of the survey is not included and should be provided.

We added this information in the methods section.

Line 230 (patients): “Answering the survey is estimated to take about 30 minutes.”

Line 288-289 (informal caregivers): “Answering the survey is estimated to take about 15 minutes.”

---

## [Decision Letter · Decision Letter 2]

10 Dec 2024

Care for post-COVID-19 condition in Germany from the perspectives of patients, informal caregivers and general practitioners: Study protocol for a mixed methods study

PONE-D-24-12578R2

Dear Dr. Stolz,

We’re pleased to inform you that your manuscript has been judged scientifically suitable for publication and will be formally accepted for publication once it meets all outstanding technical requirements.

Kind regards,

Nuru Abdu, BPharm

Academic Editor

PLOS ONE

Additional Editor Comments (optional):

Reviewers' comments:

Reviewer's Responses to Questions

**Comments to the Author**

1. Does the manuscript provide a valid rationale for the proposed study, with clearly identified and justified research questions?

Reviewer #1: Yes

Reviewer #2: Yes

2. Is the protocol technically sound and planned in a manner that will lead to a meaningful outcome and allow testing the stated hypotheses?

Reviewer #1: Yes

Reviewer #2: Yes

3. Is the methodology feasible and described in sufficient detail to allow the work to be replicable?

Reviewer #1: Yes

Reviewer #2: Yes

4. Have the authors described where all data underlying the findings will be made available when the study is complete?

Reviewer #1: Yes

Reviewer #2: Yes

5. Is the manuscript presented in an intelligible fashion and written in standard English?

Reviewer #1: Yes

Reviewer #2: Yes

6. Review Comments to the Author

You may also provide optional suggestions and comments to authors that they might find helpful in planning their study.

Reviewer #1: the authors have already address the previous feedback i provided. I have reviewed the feedback provided by the other reviewer and feel they have also been addressed

Reviewer #2: I have no further comments for the provided reviewed manuscript. All the comments have been addressed as suggested by the reviewers.

7. PLOS authors have the option to publish the peer review history of their article (what does this mean?). If published, this will include your full peer review and any attached files.

Reviewer #1: No

Reviewer #2: No

---

## [Editor Report · Acceptance letter]

17 Dec 2024

PONE-D-24-12578R2 

PLOS ONE

Dear Dr. Stolz, 

I'm pleased to inform you that your manuscript has been deemed suitable for publication in PLOS ONE. Congratulations! Your manuscript is now being handed over to our production team.

Kind regards, 

on behalf of

Mr. Nuru Abdu 

Academic Editor

PLOS ONE